REGISTERED REPORT PROTOCOL

# 'Sustainable Choices' - An intervention to promote climate change mitigation behaviours: Study protocol for a randomized controlled trial

**Fabian Lenhard**[1]*, **Lorena Fernández de la Cruz**[1], **Francesco Fuso Nerini**[2,3], **Katarina Axelsson**[4], **David Mataix-Cols**[1]

**1** Center for Psychiatry Research, Department of Clinical Neuroscience, Karolinska Institutet & Stockholm Health Care Services, Region Stockholm, Stockholm, Sweden, **2** Climate Action Centre, KTH Royal Institute of Technology, Stockholm, Sweden, **3** Environmental Change Institute, University of Oxford, Oxford, England, **4** Stockholm Environment Institute, Stockholm, Sweden

☯ These authors contributed equally to this work.
* Fabian.lenhard@ki.se

## Abstract

### Introduction

The latest climate reports underscore the importance of climate change mitigation behaviours in reducing greenhouse gas emissions and achieving the climate goals. There is a growing demand among the general population for guidance on sustainable behaviour change. Previous research has indicated that behavioural interventions could play a significant role in facilitating this type of behaviour change. However, the effectiveness of such approaches has yet to be rigorously tested under scientific conditions. This project aims to assess whether a novel, intervention "Sustainable Choices" effectively promotes climate change mitigation behaviours when compared to a control condition.

### Methods

Voluntary participants will be recruited from the general population in Sweden and randomized to either "Sustainable Choices", an online, behavioural intervention targeted at Climate Change Mitigation Behaviours (CCMBs), or a waitlist, both 5 weeks of duration. The target sample size is N = 680. The primary outcome of the study will be pre- to post-intervention difference in climate change mitigation behaviours, measured with the Climate Mitigation Behaviour Scale (CLIMBS). Secondary outcomes will include acceptability measures and effects on psychological well-being. Naturalistic follow-up assessments will be administered at 1-, 3- and 6-months after the post-measurement.

**Data availability statement:** Relevant data from this study will be made available as a reduced and anonymised data set upon study completion.

**Funding:** This work was funded by Karolinska Institutet's research foundations grant 2024-2025. The funder had no role in study design, data collection and analysis, decision to publish, or preparation of the manuscript.

**Competing interests:** The authors have declared that no competing interests exist.

## Discussion

The increasing awareness of the link between climate change and individual sustainable lifestyle choices has generated a significant interest in this field. Our aim is to evaluate a scalable intervention targeted at CCMBs, while also promoting psychological well-being associated with sustainable lifestyles.

## Introduction

Scientists agree that human activity is the main cause of ongoing global warming, and that rapid and significant action is needed to reduce emissions and mitigate the changes that are already occurring [1]. Climate change threatens eco-systems as well as human health and welfare, and undermines the achievement of the Sustainable Development Goals [2]. The European Union (EU)'s member states have agreed to target climate change by reducing greenhouse gas (GHG) emissions by 55% by 2030 and reach net-zero emissions by 2050 [3]. On a per-capita perspective, emissions should drop to 1 metric ton dioxide equivalents ($CO_2e$) per-capita between now and 2050 [4], in line with goals defined by the Paris Agreement [5].

### Climate change mitigation behaviours

The Intergovernmental Panel on Climate Change (IPCC) states that systemic changes as well as behavioural changes are needed to meet emission targets and mitigate climate change [1]. It has been shown that accessible, low-cost, climate change mitigation behaviours (CCMBs) could make an immediate and meaningful contribution to this goal and reduce annual per-capita GHG emissions by at least 6–16% [1,6]. CCMBs include, but are not limited to [6]: food choices (e.g., shifting towards plant-based diets), mobility choices (e.g., choosing public transport over car driving or flying), and household energy management (e.g., optimizing residential heating, electricity, and water usage, installation of rooftop solar panels). When taking a broader perspective on individual-level actions and including adoption of new technologies, as well as policy support, the IPCC estimates that CCMBs could reduce GHG emissions globally by 40–70% by 2050 [1].

Large-scale international surveys show that there is a demand in the population for guidance regarding individual CCMBs: Nine out of ten European citizens believe that climate change is a serious problem and agree that GHG emissions should be reduced to a minimum by 2050 [7]. One third of EU citizens feel that they have a responsibility to make sustainable changes in their everyday lives [7]. Two thirds of Americans agree either "strongly" (18%) or "somewhat" (43%) that they feel a personal sense of responsibility to help reduce global warming [8].

### Willingness to act

Repeated international surveys by the Yale Program on Climate Change Communication assess public opinion, knowledge, and behaviours related to climate change. According to their research, the majority of the public in most countries expresses

concern about climate change and are willing to take action, but many (46%) feel unsure about what steps to take [9,10]. A representative Swedish survey studied barriers for CCMBs and found that 7 out of 10 respondents believed that their habitual behaviours constituted an obstacle [11]. This indicates that there is a demand for effective actions that individuals can take to contribute to climate change mitigation as well as a need for targeted interventions that facilitate behaviour change.

### Behavioural change interventions

Behavioural change interventions apply a range of well-established techniques, including education, planning, goal setting, reward systems, and problem solving, aimed towards modification of a specific behaviour [12]. There is meta-analytic support that such interventions effectively promote sustainable behaviours with, on average, moderate effect sizes [13,14]. Moreover, recent developments in internet-based interventions for mental health conditions have demonstrated that digital technology has the potential to leverage effective behavioural interventions at scale [15]. However, internet-based methods to deliver interventions that promote climate mitigation behaviours have not yet been tested.

Interestingly, sustainable behaviours have also been found to promote psychological well-being, as suggested by longitudinal observational studies. For example, data from the UK Household Longitudinal Study including 9 000 nationally representative households and over 22 000 individuals during two consecutive years found that household pro-environmental behaviours were predictive of life satisfaction [16]. The link between sustainable behaviours and well-being has also been established cross-culturally in a representative study with data from Brazil, China, Denmark, India, Poland, South Africa, and the United Kingdom (N = 6 969), suggesting that sustainable behaviours and well-being were associated uniformly across all seven countries [17].

### Aims

Considering the available evidence, it seems timely to develop and rigorously test a scalable, internet-delivered intervention targeting individual CCMBs. Our interdisciplinary team has developed an intervention, called 'Sustainable Choices', which innovatively combines well-established behavioural change principles and climate science. If 'Sustainable Choices' is proven effective at increasing CCMBs, it could be made broadly accessible to the public and thus contribute to meaningful reductions in GHG emissions, facilitate sustainable lifestyles, and improve the overall well-being of individuals on a larger scale. The primary aim of the project is to test whether 'Sustainable Choices' is effective in promoting CCMBs, compared to the control group. Secondary aims include exploring whether the intervention is effective in promoting psychological well-being, compared to the control group, whether the intervention acceptable for participants, and whether the effects of the intervention are maintained up to 6 months after the end of the intervention.

## Methods

### Study design

The design of this study is a RCT where participants are allocated (1:1) to an intervention, 'Sustainable Choices', or a waitlist control condition for a period of 5 weeks. The primary endpoint for analyses at post-intervention (after 5 weeks). Participants in the control condition are offered to cross-over to the intervention after the primary endpoint. Naturalistic follow-up assessments will be performed 1, 3, and 6 months after the primary endpoint. The trial results will be reported according to CONSORT standards for RCTs [18]. Fig 1 describes the flow of participants with all assessment points.

### Participants

Participants are recruited nationally via social media from the general population in Sweden. Interested individuals can register online and answer a series of screening questions. Inclusion criteria are A) 18 years of age or older, B) living in

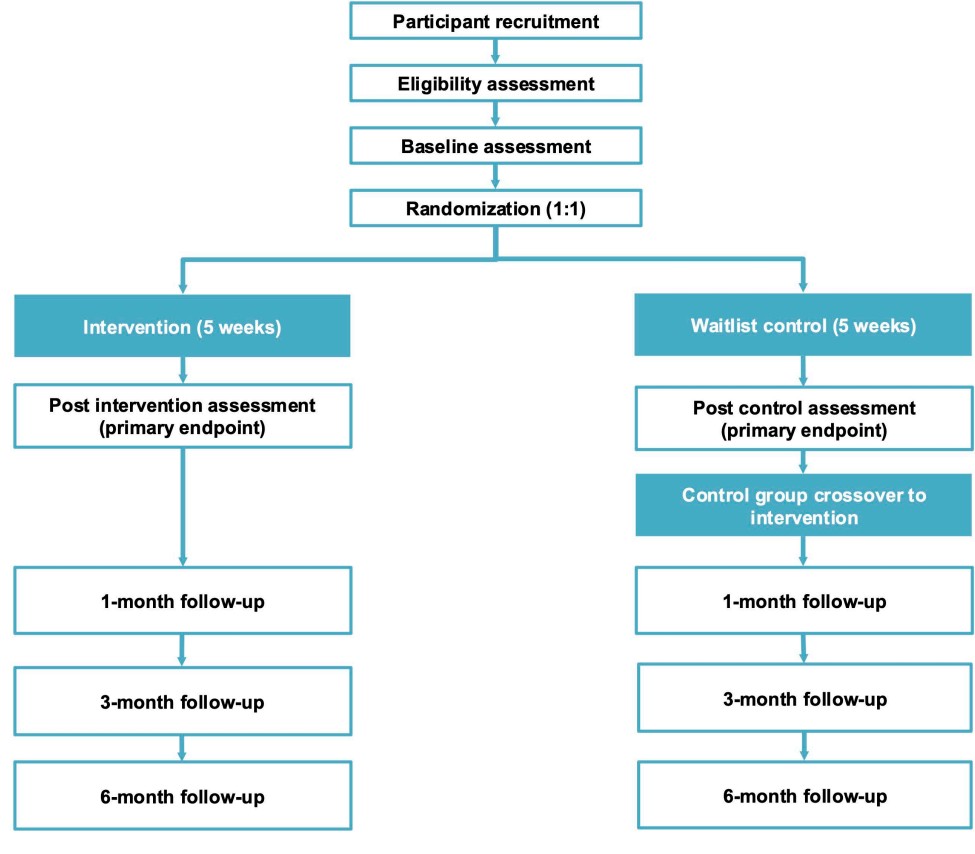

**Fig 1. Study flow chart.**

Sweden, C) at least moderate interest in sustainable behaviours (self-rated), and D) able to assign at least one hour per week to the intervention (self-rated). Exclusion criteria are A) lack of ability to read and write in Swedish or B) no regular access to the internet. Eligible participants will then proceed to complete the baseline measures. All participants are required to provide written informed study consent prior to inclusion.

The following variables will be collected to characterize the participant study sample:

The Short-Form Six Audiences of Global Warming is a four-item questionnaire based on research by the Yale Program on Climate Change Communication, categorizing the public into six distinct groups based on their beliefs and concern about climate change [19].

In addition, participants are asked about their age, sex, level of education, geographic region, and residential area (rural area, small urban area, city/larger urban area).

## Interventions

**Intervention "Sustainable Choices".** The intervention "Sustainable Choices" has been developed based on knowledge from the climate change mitigation field [1,6,20], behavioural science [12,21], and internet intervention research [21]. The format of the intervention will be similar to a self-paced, online course and consist of texts, pictures and videos, and practical exercises. The content of the intervention is presented online via personal log in, giving access to five modules over five weeks. See Table 1 for a content description of the intervention.

**Table 1. Content description of the online course "Sustainable Choices".**

| Module | Content | Behavioural change strategies |
|---|---|---|
| 1) Foundations and core values | This module covers foundational information on climate change and the available mitigation strategies. The participant reflects over individual core values. | Information<br>Reflection |
| 2) Lifestyle | The participant learns more about the GHG emissions connected to the personal lifestyle, high impact choices, and available alternatives. | Information<br>Reflection<br>Goal setting |
| 3) Social networks | This module covers social aspects of climate change mitigation such as talking with others about climate change and reinforcing constructive behaviours. | Information<br>Reflection<br>Goal setting |
| 4) Organizational and democratic processes | The module covers the importance of organizational and democratic processes as pathways for facilitation of sustainable development. It includes workplace-related behaviours as well as strategies for advocacy. | Information<br>Reflection<br>Goal setting |
| 5) Long-term behavioural strategy | In this module, the participant puts together tools from the previous modules and develops an individual long-term strategy. | Goal setting<br>Problem solving<br>Long-term planning<br>Social support |

The intervention is hosted by the KI eHealth Core Facility's technical platform "BASS" (https://ki.se/en/research/ehealth-core-facility).

**Waitlist control condition.** Participants in the waitlist group will receive a login to the online platform to access a short information page with explanatory information on the timeline of the project and the remaining waiting time for each individual.

## Outcomes

All outcome measures are remotely administered via an online service, which ensures automatic and complete entry of each measure into the trial database. Measurements will be at baseline (week 0), post-intervention (after 5 weeks, primary endpoint), and 1, 3, and 6 months after the intervention (see Fig 1).

**Primary outcome.** Climate mitigation behaviour will be assessed using the Climate Mitigation Behaviour Scale (CLIMBS), a 23-item instrument developed to capture everyday behaviours directly related to greenhouse gas mitigation [22]. Psychometric evaluation in the baseline sample supported a bifactor structure estimated using a graded response item response theory (IRT) model [22]. The best-fitting solution comprised one general factor reflecting overall mitigation propensity and four orthogonal domain factors: Mobility, Food, Consumption, and Engagement. Model fit indices indicated good fit, and reliability analyses supported a strong general dimension ($\omega\_total \approx .83$; $\omega\_H \approx .69$).

In line with this structure, the primary outcome will be the general CLIMBS latent factor score, estimated using Expected A Posteriori (EAP) scoring from the bifactor graded response model. Domain scores will be computed and analysed as secondary outcomes. The graded response framework accommodates the ordered categorical response formats used across items (frequency, counts, and stage-of-adoption anchors) while allowing estimation of a shared latent structure.

The primary outcome will be the general CLIMBS latent factor (EAP score), reflecting overall climate mitigation behaviour. Domain scores (Mobility, Food, Consumption, Engagement) are treated as secondary/exploratory outcomes, given their coexistence with a strong general factor.

**Secondary outcomes.** *Psychological well-being* is measured using three measures, capturing general well-being, climate change-related distress, and sense of meaning and purpose.

*General well-being* is measured using the Short Warwick-Edinburgh Mental Wellbeing Scale (SWEMWBS) [23], a 7-item self-report scale that has been widely used within intervention science [24]. Higher scores indicate better subjective wellbeing.

*Climate change-related distress* is measured using two validated, brief measures for low mood (PHQ-2) [25] and worry (GAD-2) [26], adapted following previous research [27]. Both the GAD-2 and PHQ-2 cover the last 2-week period. The adapted GAD-2 includes two questions regarding anxious feelings: "Feeling nervous, anxious, or on the edge because of climate change" and "Not being able to stop or control worrying about climate change". The PHQ-2 includes two questions regarding "Little interest or pleasure in doing things because of climate change" and "Feeling down, depressed, or hopeless because of climate change". Higher scores indicate more climate change-related low mood and worry, respectively.

*Meaning and purpose* will be assessed with the NIH-PROMIS Meaning and Purpose—Short Form 6a [28]. This 6-item instrument is part of the Patient-Reported Outcomes Measurement Information System (PROMIS), led by the National Institutes of Health (NIH), and evaluates perceived sense of meaning and purpose in life. Each item is rated on a 5-point Likert scale with responses ranging from "strongly disagree" to "strongly agree" and from "not at all" to "very much." Higher scores indicate greater levels of meaning and purpose. Raw scores will be converted to standardized T-scores (mean = 50, SD = 10) based on the PROMIS scoring guidelines, enabling comparison to normative reference populations [28].

**Acceptability outcomes** include recruitment rate (number of weekly participant inclusions), participant retention (number of completed intervention modules), and participants' overall satisfaction with the intervention (using a satisfaction rating for each module). Acceptability outcomes will only be collected once, either during recruitment (recruitment rate, inclusions), during the intervention (retention) or at the post-intervention assessment (satisfaction).

## Sample size

The sample size was determined to detect a between-group difference in the primary outcome (general CLIMBS latent factor score) using a two-sided test of mean differences ($\alpha$ = .05, power = 90%). Assuming a standardized mean difference of g = 0.3 [13], approximately 260 participants per arm are required. Inflating this by 30% to account for expected attrition in behavioural trials resulted in a recruitment target of approximately 680 participants. Sensitivity analyses indicate adequate power for effects in the range g = 0.30–0.40.

## Randomization

Participants are randomly assigned on a 1:1 ratio to either arm (intervention or control). The randomization sequence is created through a digital randomization algorithm (random.org) and handled by an external administrator outside of the research team to ensure integrity of the randomization procedure.

## Blinding

All outcome measures in this trial will be based on self-reported questionnaires completed directly by participants. As no external assessors will be involved in the data collection, blinding of outcome assessors is not applicable.

## Statistical methods

The principal analyses will follow an intent-to-treat statistical analysis plan.

Patterns and extent of missing data will be examined descriptively, including proportions of missingness per variable and associations with observed participant characteristics. CLIMBS item-level missingness will be handled directly within the graded response IRT framework using marginal maximum likelihood estimation, which accommodates incomplete response patterns under the assumption of missing at random (MAR). No imputation of item responses is planned. If missingness in covariates or outcome data exceeds trivial levels, appropriate model-based methods consistent with the assumed missingness mechanism will be applied.

CLIMBS will be analysed using a bifactor graded response IRT model to obtain a general mitigation behaviour score on a common metric across trial arms and assessment waves. Item parameters will be estimated jointly (not separately by arm). Intervention effects will be tested using mixed-effects models with participant-level random intercepts (or equivalent marginal approaches) to account for within-person correlation over repeated assessments. The primary analyses will use the general factor CLIMBS score as the dependent variable. Sensitivity analyses using the unit-weighted total score will be conducted to assess robustness of findings. All randomized participants will be included in regression analyses, with separate analyses for all continuous primary and secondary outcomes and time and group as predictor variables. Between-group effect sizes at the primary endpoint will be estimated using Cohen's *d*. In addition, a Complier Average Causal Effect (CACE) analysis will be conducted to estimate the causal effect of the intervention among participants who complied with their assignment ("compliers"). The original randomization will be used as an instrumental variable to account for potential bias due to selective participation. The CACE will be estimated using two-stage least squares (2SLS) regression with the CLIMBS general factor as the outcome. This analysis will be reported as a secondary, complementary analysis. The naturalistic follow-up (1, 3 and 6 months) will be analysed as within-group regression analyses in the intervention group, to test for maintenance of intervention effects on all outcome measures.

## Study status and timeline

Ethical approval for the study has been obtained from the Swedish Ethical Review Authority (reference number 2025-03308-01). The first version of this study protocol was submitted in August 2025. Participant recruitment started on 1st of September 2025 and was completed 19th of September 2025 (N = 694). Data collection is scheduled to be completed in April 2026.

## Discussion

This project addresses climate change mitigation by focusing on individual-level behavioural change, while also recognizing its connection to broader societal transformation processes. Amid increasingly urgent climate challenges, it is essential to explore not only policy- and technology-driven solutions, but also strategies that empower individuals to take meaningful action. Recent large-scale surveys across Europe and North America consistently show that the general public express high levels of concern about climate change and a growing desire to engage in mitigation efforts [7,8]. These findings suggest a clear demand for practical guidance on how to adopt more sustainable lifestyles.

The goal of the current project is to develop and evaluate the effectiveness of a brief, internet-based intervention—"Sustainable Choices" —designed to promote CCMBs. A key strength of this approach is its scalability: the intervention is entirely self-guided and delivered online, making it accessible to a wide audience without the need for specialized personnel or infrastructure.

The "Sustainable Choices" project seeks to apply behavioural science principles to climate action. Importantly, the intervention is designed to prioritize high-impact behavioural domains—such as food choices, travel, energy use, and consumption—while also attending to individual psychological well-being. This dual focus reflects recent research suggesting that sustainable behaviours can, under the right conditions, co-benefit personal well-being, rather than burden it [16].

## Limitations

Several limitations apply to this study. Firstly, the choice of a waitlist condition does not allow for control of expectancy effects or unspecific intervention effects, such as exposure to novel content. Future evaluations should therefore consider study designs with more rigorous control conditions. Secondly, the current study uses a sample of voluntary individuals, most likely motivated for behavioural change. This recruitment strategy may introduce a selection bias in the sample and limits therefore generalizability to the broader public. Future evaluations of the 'Sustainable Choices' intervention should involve representative population samples. Thirdly, the follow-up period is limited to a half-year period, and thus long-term effects beyond this time frame will not be captured in the current study. Lastly, the current study makes use of self-report

measures to capture primary and secondary outcomes. While there is some evidence that self-rating is associated with observable behaviours [29–31], the results warrant validation through study designs using objective measures.

## Conclusion and further directions

This study has potential to contribute to the emerging field of climate change mitigation through behaviour change by offering a model for scalable, evidence-based intervention. By combining rigorous evaluation with an interdisciplinary scope, it aims to support individuals in making sustainable lifestyle changes that are both environmentally impactful and psychologically beneficial. If successful, a next step to further improve the intervention could involve the integration of artificial intelligence to provide feedback on individual lifestyle profiles and suggest personalized behavioural choices with high impact on climate change mitigation. Also, further evaluations in different populations, languages and cultures would be important if the current study provides encouraging results.

## Author contributions

**Conceptualization:** Fabian Lenhard, Lorena Fernández de la Cruz, Francesco Fuso Nerini, Katarina Axelsson, David Mataix-Cols.

**Data curation:** Fabian Lenhard.

**Formal analysis:** Fabian Lenhard.

**Funding acquisition:** Fabian Lenhard.

**Investigation:** Fabian Lenhard.

**Methodology:** Fabian Lenhard, Lorena Fernández de la Cruz, Francesco Fuso Nerini, Katarina Axelsson, David Mataix-Cols.

**Project administration:** Fabian Lenhard.

**Resources:** Fabian Lenhard.

**Supervision:** David Mataix-Cols.

**Visualization:** Fabian Lenhard.

**Writing – original draft:** Fabian Lenhard, Lorena Fernández de la Cruz, Francesco Fuso Nerini, Katarina Axelsson, David Mataix-Cols.

**Writing – review & editing:** Fabian Lenhard, Lorena Fernández de la Cruz, Francesco Fuso Nerini, Katarina Axelsson, David Mataix-Cols.

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
