## [Decision Letter · Decision Letter 0]

11 Feb 2026

Dear Dr. Lenhard,

Thank you for submitting your manuscript to PLOS ONE. After careful consideration, we feel that it has merit but does not fully meet PLOS ONE’s publication criteria as it currently stands. Therefore, we invite you to submit a revised version of the manuscript that addresses the points raised during the review process.

**ACADEMIC EDITOR: Please address the comments raised by the revewers, particularly those concerning the statistics and methods.**

We look forward to receiving your revised manuscript.

Kind regards,

Laura Brunelli, MD, PhD

Academic Editor

PLOS One

Journal Requirements:

2. In your cover letter, please confirm that the research you have described in your manuscript, including participant recruitment, data collection, modification, or processing, has not started and will not start until after your paper has been accepted to the journal (assuming data need to be collected or participants recruited specifically for your study). In order to proceed with your submission, you must provide confirmation.

[This work was funded by Karolinska Institutet's research foundations grant 2024-2025.].

Reviewers' comments:

Reviewer's Responses to Questions

**Comments to the Author**

1. Does the manuscript provide a valid rationale for the proposed study, with clearly identified and justified research questions?

Reviewer #1: Yes

Reviewer #2: Yes

2. Is the protocol technically sound and planned in a manner that will lead to a meaningful outcome and allow testing the stated hypotheses?

Reviewer #1: Yes

Reviewer #2: Partly

3. Is the methodology feasible and described in sufficient detail to allow the work to be replicable?

Reviewer #1: Yes

Reviewer #2: No

4. Have the authors described where all data underlying the findings will be made available when the study is complete?

Reviewer #1: Yes

Reviewer #2: Yes

5. Is the manuscript presented in an intelligible fashion and written in standard English?

Reviewer #1: Yes

Reviewer #2: Yes

You may also provide optional suggestions and comments to authors that they might find helpful in planning their study.

Reviewer #1: The study protocol is presented in detail and the subject of the research is timely and relevant. The study will be carried out in a single country, but there is potential for future similar studies in other countries/geographical contexts that may provide an important body of research and robust data about the topic of climate change mitigation behaviors.

Some data restriction will be applied but this choice is justified in the text (264-266), as an anonymized and reduced dataset will be made available for replication of the main results.

I agree that choosing an entirely self-guided intervention is a strength of this study, as this may provide further evidence about the fact that any person should be allowed to find his/her own way to adopt sustainable behaviors and choices, adapting them into their own daily context, leading to stronger and enduring good habits and behaviors that improve climate change mitigation by leveraging the action of as many people as possible.

I also agree that using a sample of voluntary individuals, probably with a high motivation for behavioral change, is a selection bias that may compromize the generalizability of the results. The authors may suggest possible solutions for future further research in order to avoid this problem, while very common in this type of studies.

In the conclusion section (307) there is a minor mistake (I think "the integration artificial intelligence" was "the integration of artificial intelligence").

Please check again the text to be sure to use British or American spelling consinstently.

The protocol is overall very well presented and I think is suitable for publication.

Reviewer #2: The manuscript proposes a randomised controlled trial protocol to evaluate the ``Sustainable Choices" online intervention to promote climate change mitigation behaviours. The protocol is well motivated, and its overall discussion is satisfying. Nevertheless, the statistical methodology section is not exhaustive, and some gaps must be addressed before it can be considered appropriate.

In this regard, please see the attached report.

**Do you want your identity to be public for this peer review?** For information about this choice, including consent withdrawal, please see our For information about this choice, including consent withdrawal, please see our Privacy Policy .

Reviewer #1: No

Reviewer #2: No

---

## [Author Response · Author response to Decision Letter 1]

2 Mar 2026

Dear Editor,

thank you for the opportunity to revise this study protocol. All relevant information regarding the revision process are available in the cover letter and the point-to-point responses to the reviewers.

Best regards,

Dr Fabian Lenhard, on behalf of all coauthors

---

## [Decision Letter · Decision Letter 1]

17 Mar 2026

‘Sustainable Choices’ - An intervention to promote climate change mitigation behaviours: Study protocol for a randomized controlled trial

PONE-D-25-45042R1

Dear Dr. Lenhard,

We’re pleased to inform you that your manuscript has been judged scientifically suitable for publication and will be formally accepted for publication once it meets all outstanding technical requirements.

Kind regards,

Laura Brunelli, MD, PhD

Academic Editor

PLOS One

Additional Editor Comments (optional):

Reviewers' comments:

Reviewer's Responses to Questions

**Comments to the Author**

1. Does the manuscript provide a valid rationale for the proposed study, with clearly identified and justified research questions?

Reviewer #2: Yes

2. Is the protocol technically sound and planned in a manner that will lead to a meaningful outcome and allow testing the stated hypotheses?

Reviewer #2: Yes

3. Is the methodology feasible and described in sufficient detail to allow the work to be replicable?

Reviewer #2: Yes

4. Have the authors described where all data underlying the findings will be made available when the study is complete?

Reviewer #2: Yes

5. Is the manuscript presented in an intelligible fashion and written in standard English?

Reviewer #2: Yes

You may also provide optional suggestions and comments to authors that they might find helpful in planning their study.

Reviewer #2: I thank the authors for the thorough revision. They have clearly addressed the previous concerns, and the improvements are substantial. The paper is now much stronger and, in my view, ready for acceptance.

**Do you want your identity to be public for this peer review?** For information about this choice, including consent withdrawal, please see our For information about this choice, including consent withdrawal, please see our Privacy Policy .

Reviewer #2: No

---

## [Editor Report · Acceptance letter]

PONE-D-25-45042R1

PLOS One

Dear Dr. Lenhard,

I'm pleased to inform you that your manuscript has been deemed suitable for publication in PLOS One. Congratulations! Your manuscript is now being handed over to our production team.

Kind regards,

on behalf of

Dr. Laura Brunelli

Academic Editor

PLOS One